# Association of Ct Values from Real-Time PCR with Culture in Microbiological Clearance Samples for Shiga Toxin-Producing *Escherichia coli* (STEC)

**DOI:** 10.3390/microorganisms8111801

**Published:** 2020-11-16

**Authors:** Michael Bording-Jorgensen, Brendon D. Parsons, Gillian A.M. Tarr, Binal Shah-Gandhi, Colin Lloyd, Linda Chui

**Affiliations:** 1Department of Laboratory Medicine and Pathology, University of Alberta, Edmonton, AB T6G 2R3, Canada; bordingj@ualberta.ca (M.B.-J.); binalvivekgandhi@gmail.com (B.S.-G.); cdlloyd@ualberta.ca (C.L.); 2Department of Microbiology and Immunology, Dalhousie University, Halifax, NS B3H 4R2, Canada; BrendonP@dal.ca; 3Division of Environmental Health Sciences, University of Minnesota, Minneapolis, MN 55455, USA; gtarr@umn.edu; 4Alberta Precision Laboratories-Public Health Laboratory (ProvLab), Edmonton, AB T6G 2J2, Canada

**Keywords:** STEC, shedding/clearance, real-time PCR, CIDT

## Abstract

Shiga toxin-producing *Escherichia coli* (STEC) are associated with acute gastroenteritis worldwide, which induces a high economic burden on both healthcare and individuals. Culture-independent diagnostic tests (CIDT) in frontline microbiology laboratories have been implemented in Alberta since 2019. The objectives of this study were to determine the association between gene detection and culture positivity over time using STEC microbiological clearance samples and also to establish the frequency of specimen submission. Both *stx* genes’ amplification by real-time PCR was performed with DNA extracted from stool samples using the easyMAG system. Stools were inoculated onto chromogenic agar for culture. An association between gene detection and culture positivity was found to be independent of which *stx* gene was present. CIDT can provide rapid reporting with less hands-on time and technical expertise. However, culture is still important for surveillance and early cluster detection. In addition, stool submissions could be reduced from daily to every 3–5 days until a sample is negative by culture.

## 1. Introduction

There are approximately 4 million reported cases annually of food-borne illness in Canada, with an estimated 4000 from 30 known and another 7600 from unknown etiologies, some of which will result in hospitalization [1]. Furthermore, self-reported cases are estimated to be 19.5 million episodes, with only 9% seeking medical care, 17% of these being asked to submit a specimen for laboratory testing, and only 49% of these were compliant [2]. The average economic cost of food-borne acute gastroenteritis (AGE) is over CAD 1000 per case, CAD 514 million (British Columbia) [3], CAD 400 million (Alberta), and an estimated total Canadian cost of CAD 3.7 billion [4]. These costs include diagnostic testing, loss of productivity, hospitalization if required, medications, and clinic visits [3]. A 2008 study in British Columbia concluded that the loss of productivity and time away from paid employment were the most significant contributors to the economic costs of AGE [3].

One of the known pathogens causing AGE is shiga toxin-producing *Escherichia coli* (STEC) [1]. STEC virulence involves two shiga toxins, encoded by the *stx*_1_ and *stx*_2_ genes within the lambdoid prophages, which inhibit protein synthesis, leading to diarrhea and the potential for bloody diarrhea and hemolytic uremic syndrome (HUS) [5]. STEC can have a variety of other virulence factors for increased pathogenicity, such as *eae* and *hly*, which are localized within the locus of the enterocyte effacement pathogenicity island, as well as other virulence plasmids such as pO157 [5,6]. Pediatric populations, particularly those under the age of 5 years, are most at risk for developing complications when infected, and there are also long-term sequelae such as the possibility of developing irritable bowel syndrome even 8 years post-infection [7].

A notably virulent STEC serotype responsible for significant AGE-related morbidity is *E. coli* O157:H7, which was first recognized in 1982 in the United States, where several outbreaks occurred [7]. The primary reservoir for STEC O157:H7 is ruminants, particularly cattle, with transmission mediated through contamination from food or water, environmental exposure, or person-to-person [8]. More recently, other STEC serotypes known as the “BIG 6” (O26, O45, O103, O111, O121, and O145) [6,7] have been involved in major outbreaks as well. One such outbreak was *E. coli* O26, associated with soft cheese made from contaminated raw cow’s milk in France (May 2019), with 16 cases of paediatric HUS reported [9]. Another notable outbreak involved O121 in Canada (2016) from contaminated flour, which resulted in 29 cases and one HUS complication [10].

Individuals diagnosed with STEC in Alberta whose occupations involve food handling or childcare are required by the Alberta Health Authority to refrain from going to work due to the potential transmission risk. Such persons may return to work once two consecutive negative stool samples are confirmed by culture, taken 24 h and 48 h apart after non-diarrheal stool has resumed [11]. Similar guidelines are applied to children attending childcare facilities, as these facilities are higher risk environments due to sub-optimal hygiene conditions [8]. This can cause a significant economic impact as individuals may be required to abstain from work until they have cleared the bacterium, or in pediatric cases, parents (caregivers) may refrain from working in order to care for their sick child who is unable to return to the childcare facility. This loss of productivity and time away from paid employment, which largely accounts for the significant economic impact of STEC, is therefore highly dependent on microbiological clearance of stool samples submitted for diagnostic testing. However, there is no established frequency for the submission of stool samples for microbiological clearance, resulting in random submission dependent entirely on the submitting patient. This also creates additional expenses for processing unnecessary specimens.

Culture-independent diagnostic tests (CIDT) have been implemented in major clinical microbiology laboratories in Alberta, Canada. Amplification assays can provide shorter turn-around times for reporting, simplified workflow, high sensitivity and specificity, and limited requirement for technical expertise [12,13]. However, culture, along with molecular typing and characterization of the isolates, remains essential for early cluster detection, epidemiological investigations, and surveillance purpose. Moreover, the detection of STEC by CIDT is only presumptive of a viable STEC shed in stool and often cannot be distinguished from a non-viable STEC. As a result, this plausibly introduces a disparity in the duration for microbiological clearance of STEC detected by CIDT versus culture methods. To assess this, we established two objectives in this study: (1) to evaluate STEC gene detection by amplification assay compared to culture positivity over time using stools submitted for STEC microbiological clearance and (2) to establish the frequency of specimen submissions.

## 2. Materials and Methods

### 2.1. Patient Samples

A total of 110 stools from 14 individuals submitted to Alberta Precision Laboratories Public Health Laboratory (ProvLab) under the Medical Officer of Health for STEC clearance testing were included. Duration of sample submission was determined from the first positive stool sample submitted to ProvLab until the patient had 2 consecutive negative samples. The initial positive stool sample was identified by the frontline microbiology laboratory with subsequent submission under the direction of the Medical Officer of Health, and these were the samples included in this study. Serotyping of the first reported isolate from each patient was provided by the Public Health Agency of Canada National Microbiology Laboratory in Winnipeg, Manitoba, Canada, except for *E. coli* O157. All *E. coli* O157 isolates were serotyped locally in the frontline microbiology laboratory and also submitted to the ProvLab for confirmation by serology or PCR.

### 2.2. Detection and Characterization of STEC

STEC-positive stools received were inoculated using a 10 µL loop (Fisher Scientific, Ottawa, ON, Canada) onto ChromAGAR™ STEC (ChromAGAR Paris, France) agar upon receipt in ProvLab. After 20 to 24 h of incubation at 37 °C, plates were examined for the presence of mauve color colonies [14]. DNA from stool samples was extracted using the NucliSENS easyMag system (bioMerieux, Montreal, QC, Canada). Approximately 10 µL loopful of stool (~100 mg) was suspended in 1 mL of NucliSENS^®^ Lysis buffer in SK38 soil grinding lysis bead tubes (Luminex, Toronto, ON, Canada) and shaken on a vortex at max speed for 10 min; sample was left at room temperature for 15 min, followed by centrifugation at 15,871× *g* for 5 min. A total of 200 µL was extracted, with a final elution volume of 70 µL.

The primers and probes (Integrated DNA Technology, Skokie, IL, USA) used for amplifying the shiga toxin genes (*stx*_1_ and *stx*_2_) [15] are shown in Table 1. The total reaction contained 12.5 μL of 1X PrimeTime^®^ Gene Expression Master Mix (Integrated DNA Technology, Skokie, IL, USA), 0.33 μM of each primer, 0.22 μM probe, 5 μL DNA template, and molecular biology grade water in a total of 25 μL reaction volume. Positive (*E. coli* O157:H7 strain EDL933 DNA) and no template controls were included in each run and qPCR assays were repeated in triplicate. qPCR amplification conditions consisted of 95 °C for 1 min followed by 40 cycles of 95 °C for 5 s and 58 °C for 45 s performed on the 7500 FAST real-time PCR system (Applied Biosystems, Foster City, CA, USA). An average Ct value corresponding to the quadrant of bacterial growth observed was used for each patient sample. For plotting purposes, qPCR-negative samples were given a Ct value of 40 as this was the maximum number of cycles in addition to any sample showing a Ct above 36 being determined as negative by our laboratory.

### 2.3. Statistics

A nonparametric maximum likelihood estimate was fit to the distribution of the interval-censored clearance times using the Turnbull algorithm [16]. Intervals were defined based on the first day culture-negative or with a Ct value > 36.0 of two consecutive negative tests of the given type. Modified bootstrap 95% confidence intervals were obtained from 10,000 iterations and plotted using the interval package in R [17]. We determined whether Ct could predict quadrant using a cumulative link mixed model fitted with the Laplace approximation. Quadrant, the response variable, was modeled as ordinal with five levels (negative and Q1–Q4) and unstructured thresholds, with a logit link. Ct was modeled as a continuous variable and patient was included as a random effect to incorporate all paired culture–PCR repeated measurements. All estimation was done using the ordinal package in R [18].

## 3. Results

A total of 14 patients for microbiological clearance investigation were included in the study, with the number of submitted stools ranging from 2 to 19 samples per individual. The detection of clearance, based on the first negative sample, in this investigation was found to be an average of 18 days (median of 18 days, culture-based) or 22 days (median of 21 days, molecular-based), with the shortest being 3 days (Patient G) and the longest being 38 days (Patients D and J) (Table 2). Clearance curves based on the Turnbull algorithm (Figure 1) showed slightly longer time to clearance by PCR, with considerable overlap between the two methods. Patients A, G, I, and N were found to be PCR-positive for several days after being culture-negative. The majority of the samples were *stx*_1_-positive, with the exception of two *stx*_2_ (Patients A and K) and two *stx*_1_ and *stx*_2_ (Patients J and M). Serotypes included O157 and non-O157 STEC belonging to both “BIG 6” (O26, O103, O111 and O121) and non-top 6 (O118 and O186) (Table 2).

For our real-time PCR assay, the threshold with a Ct value of ≥36 was considered negative based on our laboratory verification results. The presence of mauve-colored colonies on CHROMagar™ STEC is indicative of STEC growth and the presence of blue colonies shows non-specific growth of other bacteria. Observation of mauve colonies in a particular quadrant of the agar plate corresponded with the Ct values of the *stx* real-time PCR assays, as shown in Appendix A. For the association between Ct and quadrant, the OR was 0.71 (95% CI 0.64, 0.79), meaning that for each one-cycle increase in the Ct, the quadrant odds fell by ~30%. This was similar when examining *stx*_1_ alone (OR 0.72; 95% CI 0.65, 0.80). The sample size was insufficient to obtain a standard error (and thus a confidence interval) for *stx*_2_ alone; the point estimate was OR 0.65 (Figure 2). These results support an association between lower Ct and STEC colonies reaching further around the plate.

Furthermore, the microbiological clearance duration depends on the individual as the duration between the first positive stool sample for clearance testing under the direction of the Medical Officer of Health to the point of obtaining a negative molecular assay varied between individuals (Table 2). The frequency of stool samples submitted not only varied between patient but also between samples, with some being daily and others up to 7 days in between submission.

## 4. Discussion

This study is important as the pediatric population, particularly children under the age of 6 years, are at higher risk for complications and more likely to transmit to the adult population via person-to-person contact [19,20]. While data for shedding are abundant for children, there are few studies looking at the adult population, who may serve as a secondary cause of infection [16]. Individuals involved in our study were referred by the health authority for microbiological clearance under the Alberta Health Act. Duration of clearance in our study (average of 18 days with a median of 18 days) was similar to what was observed during the 2011 STEC O104 outbreak in Germany, where the median faecal shedding was determined to be between 17 to 18 days [21]. Other pathogens associated with AGE have been documented with similar faecal shedding; for example, studies have found that individuals with norovirus had a median shedding of 27–28 days [22,23]. Overall, we found that *stx* Ct is predictive of culture quadrant; however, shedding also appears to be highly variable between patients. This may reflect sample submission, as patients would likely have had symptoms before submitting a sample, visited a physician, or other host factors that were not investigated in this study.

Correlating culture positivity with gene detection is particularly important with the implementation of CIDT. As molecular assays, such as CIDTs, target the pathogen’s DNA, it is possible that DNA from an organism may persist while the patient is no longer symptomatic and the organism no longer recoverable [13]. This was seen in several patients (A, G, I, and N) where PCR positivity persisted for several days after culture was negative. Thus, relying solely on molecular assays for pathogen detection in these cases may further increase the economic cost associated with refraining from work and childcare services, as well as repeat testing. Our data suggest an association between molecular assays and culture as growth on CHROMagar™ STEC. There was no difference as a result of whether the STEC involved was *stx*_1_, *stx*_2_, or had both genes. Determining culture quadrant growth can be subjective, which may explain how similar the Ct values are; therefore, we determined that for a culture to be in a particular quadrant, there needed to be complete growth. As an example, the first two images for Patient M in Figure 2B are recorded with quadrant 2 growth and not quadrant 3. The first sample received for Patient D showed no growth on CHROMagar™ but was real-time PCR-positive, which may be due to the quantity as well as the consistency of the initial stool sample as the following sample showed growth. Patient L is an interesting case where the Ct value increased with a corresponding increase in plate growth; however, it is important to note that only four samples were received for this individual and the result could be due to stool consistency and plating. Patient N had small differences in Ct value, as the first sample started out with an already high Ct value even with growth in quadrant 4. Patient I also had initial Ct values that were high, in addition to having several samples that continued to be real-time PCR- but not culture-positive (Appendix A).

To our knowledge, this is the first study to examine the relationship between culture positivity and molecular detection with patient samples as other studies have used animals [24,25]. One such study has found no correlation between gene detection of *stx* and isolation of STEC by culture from individual cows but there was for environmental detection on the cattle farm [26]. This is in contrast to our study, where isolation and gene detection correlated with the majority of samples, except in patients A, G, I, and N with positive gene detection when culture was negative. Due to the large variation in sample submission ranging from 2 to 18 days in some cases, it is possible that other patients could have been culture-negative but gene-positive earlier. The time that lapsed between sample submission was neither consistent between samples nor individuals. Patients D, J, and K are examples where we received 4, 7, and 8 samples but the duration was 45, 43, and 34 days, respectively. Patient C is an interesting case as culture positivity persisted past molecular positivity as STEC growth was negative by day 33 whereas real-time PCR was negative on day 29, 4 days earlier, with growth spreading into quadrant 2 and a Ct value of 36 (Appendix A), which is equal to our cutoff and was therefore considered real-time PCR-negative.

Shiga toxins, particularly Stx2, are noted to be important for the development of HUS and therefore it is essential to characterize the patient infectious status over time. Since the majority of frontline laboratories in Alberta have implemented CIDT as a primary screening tool, a high Ct value might have some diagnostic value for microbiological clearance in parallel with culture. We have observed this in some samples where a low Ct value also had a decrease in bacterial growth, which could be due to the fact there are dead bacteria contributing to the extracted DNA, leading to positive results by the real-time PCR assay [27]. Molecular assays targeting the genes have higher sensitivity than culture methods. A positive result can be from residual DNA from the dead bacteria in the stool, therefore not representing the presence of viable organisms [28]. This is a major drawback in clearance samples where a positive CIDT would result in the individual having to submit more samples, even though they may be culture-negative. This is shown in patient I, which was culture-negative by day 12 but was not real-time PCR-negative until day 19.

This is a pilot project with very limited samples; however, it illustrates some important points, such as the fact that stool sample collection could be changed to 3–5 days instead of daily submission to reduce diagnostic costs pertaining to materials and human resources. Furthermore, with the increasing implementation of highly sensitive molecular assays such as CIDT, the management of clearance patient samples without concurrent culture may result in more positive results due to the detection of residual DNA from non-viable bacteria. We propose that for microbiological clearance samples, a high Ct threshold can be established and samples above the threshold should be simultaneously cultured to demonstrate the detection of viable organisms. This will shorten the time for reporting via PCR and also eliminate unnecessary culture. This might be an alternative method to incorporate both CIDT and culture for efficient detection of viable organisms in stools submitted for STEC microbiological clearance in stool samples for STEC. Therefore, this study highlights the continued necessity of culture for diagnostic testing of microbiological clearance samples.

## Figures and Tables

**Figure 1 microorganisms-08-01801-f001:**
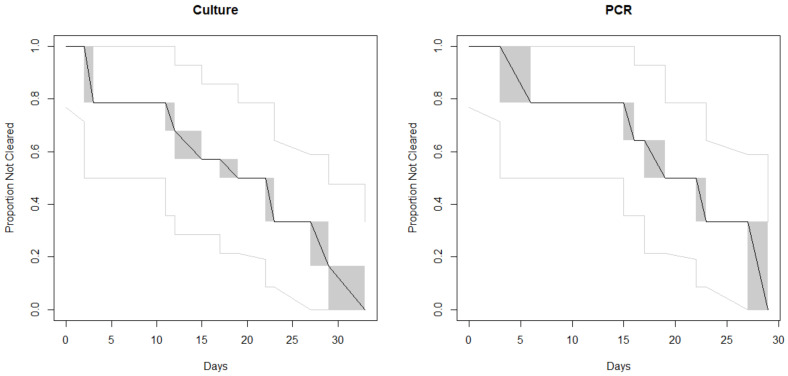
Clearance curves based on the Turnbull algorithm. The black line indicates the proportion of cases who had not yet cleared their STEC infection according to the given method, by day. The gray shaded regions are the Turnbull intervals during which clearance could have occurred. The light gray lines indicate the 95% confidence intervals.

**Figure 2 microorganisms-08-01801-f002:**
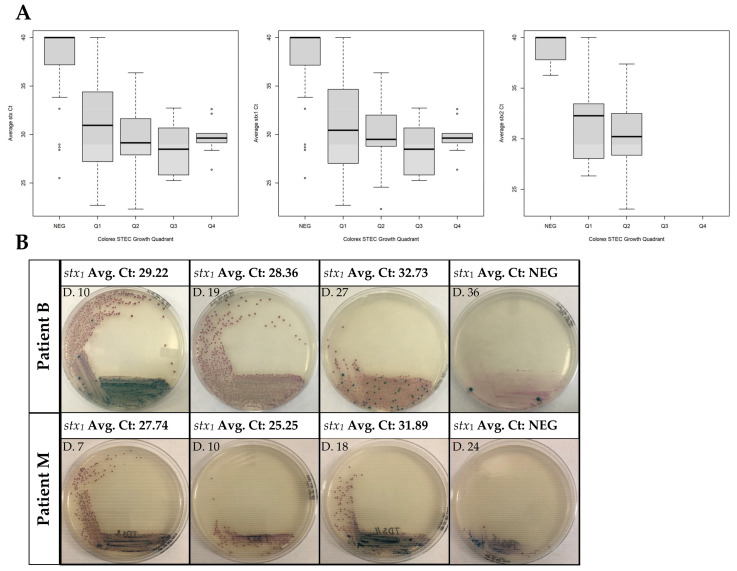
Determination of growth and presence of *stx* genes during clearance of STEC. (**A**) Boxplots showing the Ct of *stx*_1_, *stx*_2_, and lowest *stx* gene detected by quadrant for each analyzed specimen. *stx* detection was done by DNA extraction from the stool using easyMAG and analyzed with real-time PCR. The Ct value shown is a real-time PCR run from easyMAG extracted DNA, averaged from 3 independent assays. Quadrant negative (NEG) stools are defined as having no mauve colonies. Ct values were averaged from 3 independent assays. (**B**) Representative images for *stx* STEC stools. Growth determination on CHROMagar™ plates was done directly from the stool.

**Table 1 microorganisms-08-01801-t001:** Primer and probe sequences used in this study [15].

Reference Gene, Primer/Probe	Sequence 5′-3′
*stx*_1_-F	TTT GTY ACT GTS ACA GCW GAA GCY TTA CG
*stx*_1_-R	CCC CAG TTC ARW GTR AGR TCM ACR TC
*stx*_1_-P	CTG GAT GAT CTC AGT GGG CGT TCT TAT GTA A
*stx*_2_-F	TTT GTY ACT GTS ACA GCW GAA GCY TTA CG
*stx*_2_-R	CCC CAG TTC ARW GTR AGR TCM ACR TC
*stx*_2_-P	TCG TCA GGC ACT GTC TGA AAC TGC TCC
In the sequences: Y is (C, T), S is (C, G), W is (A, T), R is (A, G), M is (A, C)	

**Table 2 microorganisms-08-01801-t002:** STEC patient sample characteristics.

Patients	A	B	C	D	E	F	G	H	I	J	K	L	M	N
Number of samples submitted	4	2	10	4	12	15	6	5	10	7	8	4	19	4
Duration of sample submission (days)	19	6	49	45	48	32	10	18	20	43	34	10	31	19
Culture-negative (days)	15	6	33	38	35	26	3	16	12	38	29	9	23	17
Real-time PCR-negative (days)	19	6	29	38	35	26	6	16	19	38	29	9	23	19
*Stx* status	2	1	1	1	1	1	1	1	1	1 and 2	2	1	1 and 2	1
Serotyped	O121	O26	O111	O26	O103	O186	O118	O186	O103	O157	O157	O26	O157	O111

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
