# Peer review of "Association of Ct Values from Real-Time PCR with Culture in Microbiological Clearance Samples for Shiga Toxin-Producing Escherichia coli (STEC)"

_microorganisms, 2020, doi:10.3390/microorganisms8111801_

Round 1
Reviewer 1 Report
Type of manuscript: Article
Title: Correlation of Ct values of qPCR with Culture in Microbiological Clearance Samples for Shiga toxin-producing Escherichia coli (STEC)
Journal: Microorganisms
General Comments
With the increased use of qPCR for virus detection there needs to be some standardised tools. Different techniques are used and often the complete methodology is not reported. Hence I think there should be more publications such as this.
Specific comments
I am going to focus on the statistics. The authors seem to have combined 3 statistical analysis: linear regression (with r2 coefficient of determination), correlation with Pearson correlation coefficient (r) and concordance: a measure of agreement between 2 (for example) diagnostic tests. These are all different tests.
Determining the time of clearance should probably be estimated using interval censored nonparametric survival analysis. Interval censoring should be used as sampling does not seem to have been carried out on consecutive days. Hence all the authors know is clearance was between first negative and second negative sample (assuming 2 consecutive negatives is the criteria for clearance). This method is used in the paper that the authors cite: Duration of fecal shedding of Shiga toxin producing Escherichia coli O104:H4 in patients infected during the 2011 outbreak in Germany: a multicenter study. Clin Infect Dis, 2013. 56(8): p. 1132-40
The relationship between ct and quadrant could be analysed using a general linear mixed model with ct as the response variable and quadrant as the categorical predictor variable. It appears that quadrant is currently being treated as a continuous variable in a linear regression analysis. It may be my opinion but quadrant does not really have a numerical meaning although it is an ordered variable. All samples could be used from each patient, although patient should be included as a random effect in the model as the data from the same patient is not independent.
Author Response
Response to Reviewer 1 Comments
Pont 1: I am going to focus on the statistics. The authors seem to have combined 3 statistical analysis: linear regression (with r2coefficient of determination), correlation with Pearson correlation coefficient (r) and concordance: a measure of agreement between 2 (for example) diagnostic tests. These are all different tests.
Determining the time of clearance should probably be estimated using interval censored nonparametric survival analysis. Interval censoring should be used as sampling does not seem to have been carried out on consecutive days. Hence all the authors know is clearance was between first negative and second negative sample (assuming 2 consecutive negatives is the criteria for clearance). This method is used in the paper that the authors cite: Duration of fecal shedding of Shiga toxin producing Escherichia coli O104:H4 in patients infected during the 2011 outbreak in Germany: a multicenter study. Clin Infect Dis, 2013. 56(8): p. 1132-40
The relationship between ct and quadrant could be analyzed using a general linear mixed model with ct as the response variable and quadrant as the categorical predictor variable. It appears that quadrant is currently being treated as a continuous variable in a linear regression analysis. It may be my opinion but quadrant does not really have a numerical meaning although it is an ordered variable. All samples could be used from each patient, although patient should be included as a random effect in the model as the data from the same patient is not independent.
Response 1: We would like to thank the reviewer for their comments regarding our statistical analysis and agree that they are all different tests. Using the interval censored nonparametric survival analysis would be useful for clearance time in an epidemiological study.
Ct value is only an indication of DNA within the sample and not necessarily a reflection of viable bacteria. The main objective of this study is to illustrate the possibility of using CIDT Ct values to correlate with culture results. Based on our results, frequency of follow-up submission of microbiological samples could be reduced and therefore reduce the laboratory cost and increase the efficiency of testing.
Reviewer 2 Report
In this paper, Bording-Jorgensen et al. use qPCR of stx genes as a culture-independent diagnostic test (CIDT) and compare it to a culture-based identification method.
The authors include a thorough Introduction (with minor omissions; see specific comments below) that does an excellent job of explaining the problem they are investigating. The methods given are relatively complete, again with some minor omissions listed below. Other researchers would certainly be able to carry out the procedures using these instructions. Statistical methods are stated, but there is no mention of multiple comparison adjustment.
My biggest concern with this paper is the concept of using “quadrant growth” (i.e. how far a researcher has streaked out the sample) as a measure of bacterial load. The technique described is only barely quantitative – given that the authors are conducting statistical analysis on these data, I would have expected viable CFU counts per volume of sample as a minimum. It seems highly likely that this is one reason for the variation in results. The Discussion does a good job of analysing this issue, noting that quadrant growth is subjective and the measures taken by the authors to alleviate this where possible. However, the Discussion focuses on the subjectivity of the observations taken, rather than other sources of error such as differences in plating.
Nonetheless, the paper is interesting and attempts to answer a relevant and valuable question. I agree with the authors’ conclusions that cultures are still a necessary part of the screening process and I believe that this study does strongly support their argument.
The writing has some minor language issues, predominantly in the Abstract. There are examples of incorrect grammar, whole missing words and missing punctuation that detract somewhat from the quality of the work (see details below). Generally, however, the manuscript is well-written and easy to follow.
Specific comments:
- Line 17: “genes” requires an apostrophe, i.e. “Both stx genes’ amplification”.
- Line 19: Missing period before “Stools”.
- Lines 21-22: “CIDT can decrease turn-around-time and less labour intensive and technical expertise” needs extra words to make sense.
- Line 33: A comma is used here to separate one thousand ($1,000) whereas other times it is not (e.g. 4000, 7600). This should be made consistent.
- Line 44: Not all STEC-related virulence factors are encoded within the LEE. Whilst I appreciate that the STEC “umbrella” includes more than just O157:H7, the authors could still mention virulence plasmids such as the pO157 virulence plasmid.
- Line 95: “except for coli O157” – what happened in the case of O157? If this was only the control strain, its source should be mentioned here.
- Line 97: “was inoculated” should read “were inoculated”
- Line 110: Which strain of O157 was used for the control?
- Line 126: Did the authors adjust for multiple comparisons?
- Lines 134-135: Formatting of stx1/2 differs within this sentence. (It also differs in Figure 1.)
- Lines 135-136: Previously, these serotypes were referred to as “BIG 6” and here they are referred to as “top 6” – be consistent. (Same for Figure 1.)
- Line 140: Please provide a reference for the claims about ChromAGAR™ STEC colours.
- Line 150: Missing period before “Furthermore”.
- Line 186: “concordance results” should read “concordant results”
- Line 190: The authors refer to “Fig 2” rather than “Fig 1B”.
- The authors use “real time PCR”, “RT-PCR” and “qPCR” throughout the paper. These are not interchangeable terms, so the authors should adjust this for clarity.
Author Response
Response to Reviewer 2 Comments
Point 1: The authors include a thorough Introduction (with minor omissions; see specific comments below) that does an excellent job of explaining the problem they are investigating. The methods given are relatively complete, again with some minor omissions listed below. Other researchers would certainly be able to carry out the procedures using these instructions. Statistical methods are stated, but there is no mention of multiple comparison adjustment.
My biggest concern with this paper is the concept of using “quadrant growth” (i.e. how far a researcher has streaked out the sample) as a measure of bacterial load. The technique described is only barely quantitative – given that the authors are conducting statistical analysis on these data, I would have expected viable CFU counts per volume of sample as a minimum. It seems highly likely that this is one reason for the variation in results. The Discussion does a good job of analysing this issue, noting that quadrant growth is subjective and the measures taken by the authors to alleviate this where possible. However, the Discussion focuses on the subjectivity of the observations taken, rather than other sources of error such as differences in plating.
Response 1: We would like to thank the reviewer for their comments regarding statistical analysis and methods. We agree that quadrant growth is semi-quantitative and subjective. This was a pilot project to determine the correlation between Ct values and growth as measured by plating. This project is from a diagnostic perspective where CIDT are becoming more commonly used. We wanted to determine a general idea of what a diagnostic laboratory can use (Ct value from CIDT) for follow up sample submission before plating is required. The reviewer has a good point on potential sources of error such as plating, which we addressed in line 213 of the discussion.
Point 2:Line 17: “genes” requires an apostrophe, i.e. “Both stxgenes’ amplification”.
Response 2: Changed in revised version
Point 3: Line 19: Missing period before “Stools”.
Response 3: Changed in revised version
Point 4: Lines 21-22: “CIDT can decrease turn-around-time and less labour intensive and technical expertise” needs extra words to make sense.
Response 4: Changed in revised version
Point 5: Line 33: A comma is used here to separate one thousand ($1,000) whereas other times it is not (e.g. 4000, 7600). This should be made consistent.
Response 5: Changed to include a comma
Point 6: Line 44: Not all STEC-related virulence factors are encoded within the LEE. Whilst I appreciate that the STEC “umbrella” includes more than just O157:H7, the authors could still mention virulence plasmids such as the pO157 virulence plasmid.
Response 6: Has been included in the text now on line 45-46.
Point 7: Line 95: “except for coliO157” – what happened in the case of O157? If this was only the control strain, its source should be mentioned here.
Response 7: “Serotyping of the first reported isolate from each patient was provided by the Public Health Agency of Canada-National Microbiology Laboratory in Winnipeg, Manitoba except for coli O157”. All E. coli O157 isolates are serotyped locally in our frontline microbiology laboratory and also submitted to the provincial laboratory for confirmation by serology or PCR and they are not required to send to the National Microbiology laboratory (NML) for “O” typing. As for the other serotypes, Alberta Provincial Laboratory does the isolation of non-O157 STEC for all frontline laboratories and forward to the NML for O typing by serology or molecular methods. Clarification was added to the methods section line 96-98
Point 8: Line 97: “was inoculated” should read “were inoculated”
Response 8: Changed in revised version
Point 9: Line 110: Which strain of O157 was used for the control?
Response 9: Changed in revised version. Added that strain EDL933 was used as the positive control for the real-time PCR.
Point 10: Line 126: Did the authors adjust for multiple comparisons?
Response 10: Multiple comparisons were not done because only the relationship between Ct value and quadrant growth were being analyzed. No other predictor was being analyzed, such as serotype or stx.
Point 11: Lines 134-135: Formatting of stx1/2 differs within this sentence. (It also differs in Figure 1.)
Response 11: Changed in revised version
Point 12: Lines 135-136: Previously, these serotypes were referred to as “BIG 6” and here they are referred to as “top 6” – be consistent. (Same for Figure 1.)
Response 12: Changed in revised version to be “BIG 6”
Point 13: Line 140: Please provide a reference for the claims about ChromAGAR™ STEC colours.
Response 13: Reference was added to the Methods section (Line 102)
Point 14: Line 150: Missing period before “Furthermore”.
Response 14: Changed in revised version
Point 15: Line 186: “concordance results” should read “concordant results”
Response 15: Changed in revised version
Point 16: Line 190: The authors refer to “Fig 2” rather than “Fig 1B”.
Response 16: Changed in revised version
Point 17:The authors use “real time PCR”, “RT-PCR” and “qPCR” throughout the paper. These are not interchangeable terms, so the authors should adjust this for clarity.
Response 17: Changed in revised version to be real-time PCR
Round 2
Reviewer 1 Report
There has been no change in the statistical analysis from the previous version. The authors stress that it is a pilot study and not a formal epidemiological analysis but that is not a good excuse for poor analyses. I would rather they said that they tried to do the correct analysis but there was not enough data. Perhaps I was not clear in my review but I think they are mixing up statistical analyses. In the methods they state "Linear Regression with Pearson correlation coefficient" but Linear regression does not involve the pearson correlation coefficient. I think they need to sort out the statistics before the manuscript can be published.
Author Response
Reviewers Comments
Comment 1: There has been no change in the statistical analysis from the previous version. The authors stress that it is a pilot study and not a formal epidemiological analysis but that is not a good excuse for poor analyses. I would rather they said that they tried to do the correct analysis but there was not enough data. Perhaps I was not clear in my review but I think they are mixing up statistical analyses. In the methods they state "Linear Regression with Pearson correlation coefficient" but Linear regression does not involve the pearson correlation coefficient. I think they need to sort out the statistics before the manuscript can be published.
Response 1: We apologize for misunderstanding the reviewer’s comments in the first revision. We have brought on an epidemiologist with biostatistics training and made extensive revisions to the manuscript.
- Regarding clearance times, we agree with the reviewer that they are interval-censored. We have used the Turnbull algorithm to generate the equivalent of a Kaplan-Meier curve but for interval-censored data. As suggested by the reviewer, this is a nonparametric approach. Confidence intervals are provided to indicate the level of precision afforded by the sample size (Fig 1).
- Regarding the relationship between Ct and quadrant, we agree with the reviewer that it is appropriate to replace the simple linear regression with a generalized linear mixed model, with a random intercept for Patient, so as to make use of the repeated measures data. We have built this model with Ct as the independent variable and quadrant as the response variable (Fig 2). We believe this has two advantages over the previous parameterization and that suggested by the reviewer: 1) it better reflects the scientific question, which is whether PCR Ct can be used to predict quadrant on the culture plate, and 2) it enables accurate representation of quadrant as an ordinal variable. In accordance with this, we used a cumulative link mixed model fitted with the Laplace approximation for ordinal regression. Additional curves have been added to Figure 2.
- Changes were made to Table 2 where we have deleted the p value and r2
- Revisions were made to highlight the changes in statistical analysis in the following sections
- Abstract (lines 21-24)
- Methods (lines126-135)
- Results (lines 147-152, 162-167)
- Discussion (lines 209-211, 220-224, 228-230, 234-240)
- References were added for the statistical analysis
Round 3
Reviewer 1 Report
Dear Authors,
This manuscript has been greatly improved by the new statistical analysis. My only suggestion is that the you might consider changing the title and the aim, removing "correlation" as you have not done a correlation analysis. Perhaps "relationship (or association) of ct values of ...."
Author Response
Comment 1: This manuscript has been greatly improved by the new statistical analysis. My only suggestion is that the you might consider changing the title and the aim, removing "correlation" as you have not done a correlation analysis. Perhaps "relationship (or association) of ct values of ...."
Response 1: We would like to thank the reviewer for their suggestion and have changed the title to "Correlation of Ct values from Real-Time PCR with Culture in Microbiological Clearance Samples for Shiga toxin-producing Escherichia coli (STEC)". We have also changed the objective from correlation to association on line 3.